# Boosting Knowledge Utilization in Multimodal Large Language Models via Adaptive Logits Fusion and Attention Reallocation

**Wenbin An**[1,2,*]    **Jiahao Nie**[3,*]    **Feng Tian**[1,2,†]    **Haonan Lin**[1,2]    **Mingxiang Cai**[4]
**Yaqiang Wu**[4]    **Qianying Wang**[4,†]    **Xiaoqin Zhang**[5]    **Shijian Lu**[3,†]

[1]Xi'an Jiaotong University    [2]National Engineering Laboratory for Big Data Analytics
[3]Nanyang Technological University    [4]Lenovo Research    [5]Zhejiang University of Technology
wenbinan@stu.xjtu.edu.cn, jiahao007@e.ntu.edu.sg
fengtian@mail.xjtu.edu.cn, wangqya@lenovo.com, shijian.lu@ntu.edu.sg

## Abstract

Despite their recent progress, Multimodal Large Language Models (MLLMs) often struggle in knowledge-intensive tasks due to the limited and outdated parametric knowledge acquired during training. Multimodal Retrieval Augmented Generation addresses this issue by retrieving contextual knowledge from external databases, thereby enhancing MLLMs with expanded knowledge sources. However, existing MLLMs often fail to fully leverage the retrieved contextual knowledge for response generation. We examine representative MLLMs and identify two major causes, namely, attention bias toward different tokens and knowledge conflicts between parametric and contextual knowledge. To this end, we design *Adaptive Logits Fusion and Attention Reallocation (ALFAR)*, a training-free and plug-and-play approach that improves MLLM responses by maximizing the utility of the retrieved knowledge. Specifically, ALFAR tackles the challenges from two perspectives. First, it alleviates attention bias by adaptively shifting attention from visual tokens to relevant context tokens according to query-context relevance. Second, it decouples and weights parametric and contextual knowledge at output logits, mitigating conflicts between the two types of knowledge. As a plug-and-play method, ALFAR achieves superior performance across diverse datasets without requiring additional training or external tools. Extensive experiments over multiple MLLMs and benchmarks show that ALFAR consistently outperforms the state-of-the-art by large margins. Our code and data are available at https://github.com/Lackel/ALFAR.

## 1    Introduction

Building upon powerful Large Language Models (LLMs) [1, 2, 3, 4, 5, 6, 7, 8], Multimodal Large Language Models (MLLMs) [9, 10, 11, 12, 13, 14, 15, 16, 17] have achieved impressive performance over a wide range of vision-centric tasks such as image captioning [18, 19, 20], visual question answering [21, 22], *etc*. Nevertheless, MLLMs often struggle to handle knowledge-intensive vision-language tasks [23, 24], primarily due to the limited and outdated parametric knowledge acquired during training [25, 26]. Multimodal Retrieval Augmented Generation (MRAG) [27, 28, 29], a prevalent approach that attempts to resolve this issue, retrieves contextual knowledge from external

---

*Equal contribution

†Corresponding author

39th Conference on Neural Information Processing Systems (NeurIPS 2025).

data to empower MLLMs for accurate response generation. However, the way of exploiting the contextual knowledge remains under-explored, undermining the effectiveness of MRAG.

We examined representative MLLMs with MRAG and found that while MRAG can improve MLLM performance when high-quality contextual knowledge is retrieved (as shown in Fig. 1), MLLMs often fail to make full use of the retrieved knowledge, even when ground-truth knowledge is available. We identify two primary causes, namely, attention bias among visual and context tokens and conflicts between MLLMs' parametric knowledge and retrieved contextual knowledge. For the attention bias, MLLMs tend to allocate more attention to image tokens over context tokens, especially in shallow layers that are critical for knowledge extraction and exchange [30]. Since images often do not provide sufficient information for knowledge-intensive questions [23, 31], the attention bias hinders the effective utilization of contextual knowledge and leads to inaccurate MLLM responses. In addition, MLLMs allocate attention uniformly across context tokens without prioritization, which dilutes the contributions of query-relevant knowledge and tends to introduce inaccurate MLLM responses.

Knowledge conflicts typically arise from the discrepancy between contextual and parametric knowledge. We observe that MLLMs tend to rely excessively on their parametric knowledge even when accurate contextual knowledge is present, leading to under-utilization of contextual knowledge and counterfactual responses. Such a phenomenon is well aligned with observations in previous LLM studies [33, 34] and findings in psychology research [35, 36], both underscoring a clear preference toward intrinsic instead of retrieved knowledge. On the other end, the preference for the parametric knowledge does help when the contextual knowledge is unreliable [28, 37, 38]. This can be observed in Fig. 1, where low-quality contextual knowledge significantly degrades performance. Therefore, striking a balance between parametric and contextual knowledge while leveraging their complementary strengths is critical for generating accurate responses.

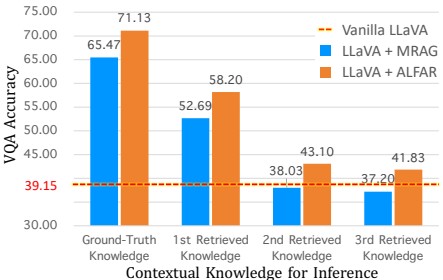

Figure 1: VQA accuracy of LLaVA-1.5 [14] on the multi-choice InfoSeek dataset [32] with respect to the quality of contextual knowledge. ALFAR fully exploits the retrieved knowledge, consistently improving MRAG performance regardless of knowledge quality.

In this work, we propose *Adaptive Logits Fusion and Attention Reallocation (ALFAR)*, a training-free and plug-and-play approach that enables effective utilization of MRAG-retrieved contextual knowledge for accurate MLLM responses. ALFAR addresses attention bias and knowledge conflicts by dynamically adjusting attention allocation and balancing the parametric and contextual knowledge, respectively. Specifically, ALFAR adaptively shifts attention from image tokens to relevant context tokens based on retrieval scores and query-context relevance, enabling MLLMs to focus on more pertinent information. In addition, ALFAR decouples parametric and contextual knowledge at output logits and weights them according to the attention distribution, enabling a balanced and synergistic integration of the two types of knowledge. Extensive experiments across multiple representative MLLMs and benchmarks demonstrate ALFAR's superior and broad applicability without involving additional training or external tools.

The contributions of this work can be summarized in three major aspects. **First**, we dive deeply into knowledge utilization in MLLMs, identifying attention bias and knowledge conflicts as two key factors that impede the effective utilization of the retrieved knowledge. These findings provide valuable insights for advancing knowledge utilization in MLLMs. **Second**, we design ALFAR, a training-free and plug-and-play approach that reallocates attention and balances parametric and contextual knowledge effectively. **Third**, Extensive experiments over multiple generative and discriminative benchmarks validate ALFAR's effectiveness and versatility, demonstrating its superior performance and broad applicability across various multimodal tasks.

## 2 Related Work

### 2.1 Multimodal Large Language Models

The rapid advancements in Large Language Models (LLMs) [1, 2, 3, 4, 5, 6, 7, 8, 39] have greatly propelled the development of Multimodal Large Language Models (MLLMs) [9, 10, 11, 13, 14, 15,

16, 40]. To align visual and textual modalities, prior studies explore different approaches such as visual encoders with linear projectors [14, 41, 40], Q-former [18, 10], and Perceivers [42], which transform image patches into visual tokens that are compatible with LLMs. Most MLLMs conduct training in two stages, namely, pre-training for feature alignment and instruction-based fine-tuning [14, 43, 10], enabling impressive performance across diverse multimodal tasks [44, 45, 46, 47]. Despite these advancements, MLLMs often face challenges in knowledge-intensive tasks [23, 31], due to the limitations of their parametric knowledge acquired during training.

### 2.2 Multimodal Retrieval Augmented Generation

Inspired by the concept of Retrieval Augmented Generation (RAG) for LLMs [48, 49], Multimodal Retrieval Augmented Generation (MRAG) has been widely explored for enhancing MLLMs with more comprehensive and up-to-date knowledge [27, 28, 29, 50]. MRAG retrieves relevant knowledge from a multimodal database and incorporates the retrieved knowledge as the context of the input. For instance, Wiki-LLaVA [27] broadens the knowledge scope of LLaVA [14] by incorporating the retrieved Wikipedia articles in training. EchoSight [29] leverages a fine-tuned Q-Former [18] to filter retrieved knowledge and enhance retrieval recall. ReflectiVA [28] and MR$^2$AG [50] introduce a trainable reflection mechanism to assess the necessity of retrieval and the relevance of retrieved knowledge. In the LLM domain, several training-free methods have been proposed to better utilize the retrieved knowledge to enhance generation quality. For instance, CAD [26] employs contrastive decoding [51] to increase the faithfulness of generation. Moreover, AdaCAD [52], Entropy [53], and COIECD [54] extend CAD [26] by introducing JS divergence, entropy, and information constraints, respectively. Despite the improved faithfulness toward the retrieved context, these methods struggle to balance parametric and contextual knowledge [55], resulting in sub-optimal performance when the contextual knowledge is noisy.

## 3 Preliminary and Motivation

### 3.1 Multimodal Retrieval Augmented Generation

Given a textual query $q$ and a query image $I$, an MLLM $\mathcal{M}_\theta$ parameterized by $\theta$ is expected to generate a reliable answer $y$. To enrich MLLMs with external knowledge, MRAG employs a multimodal retriever $\mathcal{R}_\phi$ to fetch relevant knowledge from a multimodal knowledge base $\mathcal{C} = \{(\widetilde{I}_i, c_i)\}_{i=1}^M$, where $\widetilde{I}_i$ and $c_i$ represent an image and its corresponding textual knowledge, respectively. The retriever $\mathcal{R}_\phi$ measures the similarity between the query pair $(q, I)$ and a multimodal knowledge pair $(\widetilde{I}, c)$ based on the cosine similarity between their image embeddings:

$$\alpha = \frac{\mathcal{R}_\phi(I) \cdot \mathcal{R}_\phi(\widetilde{I})}{||\mathcal{R}_\phi(I)|| \cdot ||\mathcal{R}_\phi(\widetilde{I})||} \tag{1}$$

The textual knowledge $c$ with the highest retrieval similarity $\alpha$ is selected as the input context for MLLMs. Consequently, the output distributions of the MLLM with MRAG at the time step $t$ are:

$$p(y_t) \sim \text{softmax}(\mathcal{M}_\theta(y_t|q, I, c, y_{<t})) \tag{2}$$

where $y_{<t}$ represents the sequence of generated tokens before the time step $t$.

### 3.2 Self-attention Mechanism in MLLMs

MLLMs generate responses auto-regressively using Transformer blocks [56]. Specifically, the input image, query, and context tokens are concatenated and projected into three distinct vectors: the query vector $\mathbf{Q}$, the key vector $\mathbf{K}$, and the value vector $\mathbf{V}$, through three linear layers, $W_q$, $W_k$, and $W_v$. The self-attention mechanism computes the relevance of each token to other tokens as follows:

$$\mathbf{A} = \frac{\mathbf{Q} \cdot \mathbf{K}^\top}{\sqrt{d}} + \mathbf{M} \tag{3}$$

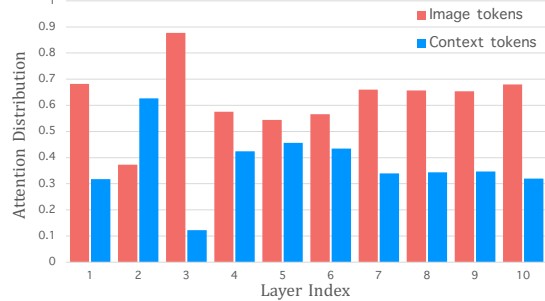

Figure 2: Proportions of attention weights that are assigned to image and context tokens at different shallow layers of LLaVA-1.5 [14].

where $\mathbf{A} \in \mathcal{R}^{n \times n}$ is the attention weight matrix, $\mathbf{M}$ is a causal mask, $d$ is the feature dimension, and $n$ is the number of input tokens. The output $\mathbf{O}$ can then be calculated by:

$$\mathbf{O} = \mathrm{softmax}(\mathbf{A}) \cdot \mathbf{V} \tag{4}$$

The attention weight matrix $\mathbf{A}$ reflects the importance of different input tokens in generating new tokens. This property makes it a valuable tool to analyze the contribution of different types of tokens to the generated responses.

### 3.3 Attention Bias in MRAG

To generate the response token $y_{n+1}$, MLLMs perform self-attention over all input tokens which correspond to the $n$-th row of the attention weight matrix $\mathbf{A}$. We analyze the contributions of the input image and context using an importance score, defined as the total attention weights assigned to these tokens. For the input image, the importance score at layer $i$ is calculated by: $S^i(I) = \sum_{j \in I} A^i_{nj}$.[1] Similarly, for the input context, the importance score at layer $i$ is determined by: $S^i(c) = \sum_{j \in c} A^i_{nj}$. As illustrated in Fig. 2, MLLMs tend to allocate more attention to image tokens than context tokens, particularly in shallow layers that are pivotal for extracting and exchanging information from distinct tokens [30]. This issue affects the effective utilization of contextual knowledge since images often do not capture sufficient information for knowledge-intensive questions [23, 31]. Moreover, MLLMs assign attention uniformly across different parts of the context without highlighting query-relevant segments. Such indiscriminate distribution increases the distraction of irrelevant knowledge, ultimately leading to inaccurate or misleading responses for MLLMs.

### 3.4 Knowledge Conflicts in MRAG

After knowledge retrieval, MLLMs integrate the retrieved contextual knowledge with their internal parametric knowledge to generate responses. However, similar to LLMs [33, 34, 52], MLLMs often encounter knowledge conflicts due to the discrepancy between the two types of knowledge, affecting the effectiveness of the model in various

Table 1: Experiments with LLaVA-1.5 [14]. Conflict Ratio: Ratio of discrepancy between parametric and contextual knowledge. Performance drop: Accuracy decline due to knowledge conflicts. Details of the two metrics are provided in Appendix A3.

|  | Infoseek | ViQuAE |
| --- | --- | --- |
| Conflict Ratio | 60.85% | 48.94% |
| Performance Drop | 28.87% | 27.02% |

practical tasks. Worse still, MLLMs tend to prioritize their parametric knowledge even when perfect contextual knowledge is provided, leading to under-utilization of contextual knowledge and factually inconsistent answers. By assuming the accessibility of the ground-truth contextual knowledge, we evaluate the conflict rate of the two types of knowledge and the resultant performance degradation on the multi-choice InfoSeek [32] and ViQuAE [24, 32] datasets with LLaVA-1.5 [14]. As shown in Tab. 1, around half of the samples exhibit knowledge conflicts, which lead to an up to 30% performance drop, highlighting the necessity of mitigating such conflicts to enhance MLLM performance on knowledge-intensive tasks.

## 4 Method

The proposed framework consists of two branches for effective handling of parametric and contextual knowledge as illustrated in Fig. 4. Within the contextual branch, we design an attention reallocation mechanism that tackles the attention bias and improves the utilization of contextual knowledge by adaptively adjusting model attention toward relevant context tokens based on query-context relevance (Sec. 4.1). In addition, the network fuses the parametric and contextual knowledge adaptively in the output logits, mitigating knowledge conflicts under the guidance of the model attention that dynamically captures the relative importance of the two types of knowledge (Sec. 4.2).

### 4.1 Attention Reallocation

As analyzed in Sec. 3.3, the attention bias results from two major factors, namely, attention preference toward image tokens and uniform attention to context tokens. We address the attention preference

---

[1]We average all attention heads and omit the notation for simplicity.

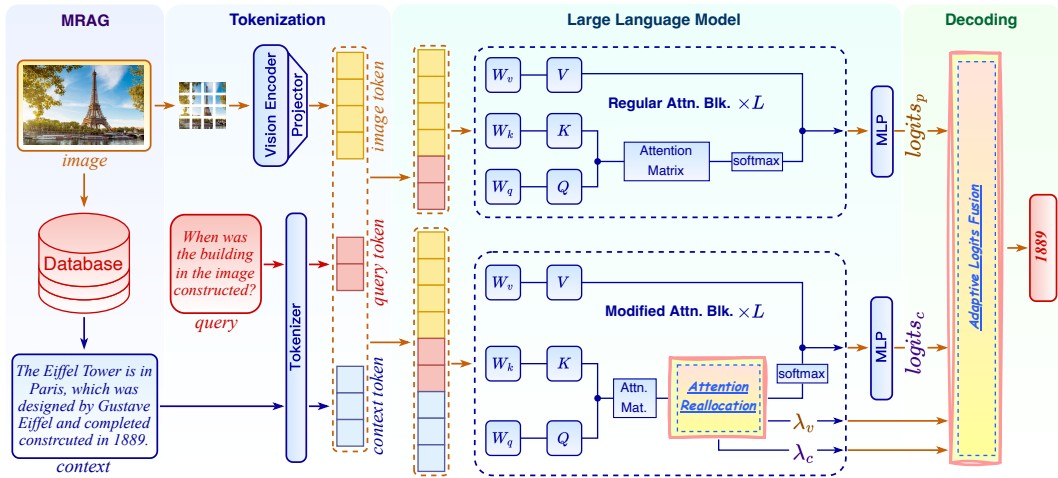

Figure 4: The overview of Adaptive Logits Fusion and Attention Reallocation (ALFAR).

by adaptively adjusting model attention as illustrated in Fig. 3, based on the retrieval similarity $\alpha$ in Eq. 1 that reflects the reliability of the retrieved context. Specifically, less attention is allocated to image tokens if the context is more reliable with a high retrieval similarity. The attention reallocation can be formulated as follows:

$$\hat{\mathbf{A}}_{ni} = (1 - \beta) \cdot \mathbf{A}_{ni}, \text{ s.t. } i \in S_I \tag{5}$$

where $\beta = k \cdot \alpha$ is the scaled retrieval similarity with a scaling factor $k$. $\hat{\mathbf{A}}$ is the modified attention weight matrix, $n$ is the number of all input tokens and $S_I$ is the index set of image tokens. $\mathbf{A}_{ni}$ corresponds to the attention weight in the $n$-th row and $i$-th column of $\mathbf{A}$.

In addition, we introduce a query-context relevance score to mitigate the uniform attention to all context tokens and allow MLLMs to focus more on query-relevant context. We derive the relevance score from attention weights assigned to query tokens by $j$-th context token as follows:

$$\omega_j = \frac{\sum\limits_{k \in S_q} \mathbf{A}_{jk}}{\sum\limits_{l \in S_c} \sum\limits_{k \in S_q} \mathbf{A}_{lk}}, \text{ s.t. } j \in S_c \tag{6}$$

where $S_q$ and $S_c$ are index sets of query and context tokens, respectively. With the relevance scores, we adaptively increase MLLMs' attention to context tokens:

$$\hat{\mathbf{A}}_{nj} = (1 + \gamma_j) \cdot \mathbf{A}_{nj}, \text{ s.t. } j \in S_c \tag{7}$$

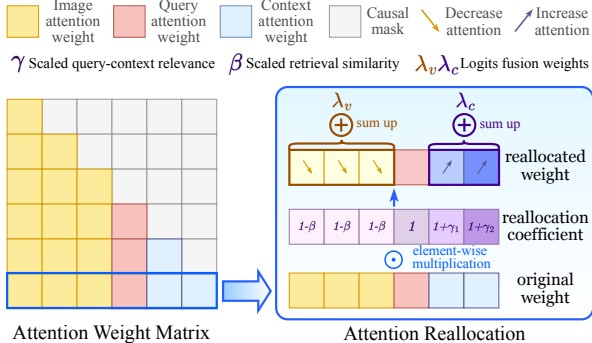

Figure 3: Attention reallocation by adjusting the last row of the attention weight matrix (i.e., $\mathbf{A_n}$).

where $\gamma_j = k \cdot \omega_j$ is the scaled query-context relevance with a scaling factor $k$. After attention reallocation, we apply softmax to redistribute the attention as in Eq. 4 to compute the output hidden states. This repeats auto-regressively for each subsequent token prediction.

## 4.2 Adaptive Knowledge Fusion

As analyzed in Sec. 3.4, parametric knowledge could hinder the utilization of contextual knowledge, leading to inaccurate responses. However, parametric knowledge brings benefits when the retrieved context is unreliable. Therefore, striking a balance between parametric and contextual knowledge based on their reliability is essential for generating accurate and reliable responses for MLLMs. Nevertheless, parametric knowledge is implicitly embedded and the two types of knowledge are entangled during inference, making it hard to explicitly represent and utilize them separately. We address this issue by disentangling the two types of knowledge and fuse them at output logits. Specifically, we represent parametric knowledge at step $t$ by using the output logits that have only query and image as inputs:

$$\text{logit}_{\text{p}} = \mathcal{M}_\theta(y_t | q, I, y_{<t}) \tag{8}$$

Table 2: VQA Accuracy comparison on generative freeform VQA datasets over three runs. *Regular* and *Parametric* denote that MLLMs generate answers with and without retrieved knowledge, respectively. The best performance is marked in **bold**.

| Model | Decoding | Human [23] | | | Validation [23] | | |
|---|---|---|---|---|---|---|---|
| | | Unseen Question | Unseen Entity | Overall | Unseen Question | Unseen Entity | Overall |
| LLaVA-1.5 | Regular [14] | $7.59_{(\pm0.08)}$ | $7.90_{(\pm0.05)}$ | $7.74_{(\pm0.01)}$ | $19.98_{(\pm0.02)}$ | $19.59_{(\pm0.01)}$ | $19.78_{(\pm0.01)}$ |
| | Parametric [14] | $6.27_{(\pm0.05)}$ | $6.26_{(\pm0.19)}$ | $6.26_{(\pm0.09)}$ | $7.14_{(\pm0.03)}$ | $6.28_{(\pm0.01)}$ | $6.68_{(\pm0.00)}$ |
| | CD [51] | $7.39_{(\pm0.21)}$ | $7.31_{(\pm0.06)}$ | $7.35_{(\pm0.09)}$ | $20.32_{(\pm0.01)}$ | $19.90_{(\pm0.00)}$ | $20.11_{(\pm0.01)}$ |
| | AdaCAD [52] | $7.81_{(\pm0.02)}$ | $8.07_{(\pm0.03)}$ | $7.94_{(\pm0.01)}$ | $21.23_{(\pm0.03)}$ | $20.91_{(\pm0.02)}$ | $21.07_{(\pm0.03)}$ |
| | Entropy [53] | $7.98_{(\pm0.09)}$ | $8.34_{(\pm0.02)}$ | $8.15_{(\pm0.03)}$ | $21.97_{(\pm0.02)}$ | $21.85_{(\pm0.01)}$ | $21.91_{(\pm0.03)}$ |
| | CAD [26] | $8.02_{(\pm0.05)}$ | $8.15_{(\pm0.03)}$ | $8.08_{(\pm0.02)}$ | $21.68_{(\pm0.01)}$ | $20.93_{(\pm0.02)}$ | $21.30_{(\pm0.02)}$ |
| | COIECD [54] | $8.78_{(\pm0.04)}$ | $8.65_{(\pm0.01)}$ | $8.71_{(\pm0.01)}$ | $22.43_{(\pm0.05)}$ | $21.73_{(\pm0.05)}$ | $22.07_{(\pm0.05)}$ |
| | AGLA [57] | $8.74_{(\pm0.28)}$ | $9.18_{(\pm0.00)}$ | $8.94_{(\pm0.08)}$ | $22.34_{(\pm0.02)}$ | $21.88_{(\pm0.01)}$ | $22.11_{(\pm0.02)}$ |
| | VCD [58] | $9.22_{(\pm0.00)}$ | $9.26_{(\pm0.00)}$ | $9.24_{(\pm0.02)}$ | $22.30_{(\pm0.03)}$ | $22.38_{(\pm0.03)}$ | $22.23_{(\pm0.03)}$ |
| | **ALFAR (ours)** | $\mathbf{12.80}_{(\pm0.09)}$ | $\mathbf{11.24}_{(\pm0.00)}$ | $\mathbf{11.96}_{(\pm0.02)}$ | $\mathbf{23.82}_{(\pm0.00)}$ | $\mathbf{23.75}_{(\pm0.01)}$ | $\mathbf{23.78}_{(\pm0.01)}$ |
| InstructBLIP | Regular [10] | $4.20_{(\pm0.00)}$ | $3.86_{(\pm0.03)}$ | $4.02_{(\pm0.01)}$ | $3.60_{(\pm0.01)}$ | $3.82_{(\pm0.00)}$ | $3.71_{(\pm0.00)}$ |
| | Parametric [10] | $4.06_{(\pm0.01)}$ | $3.65_{(\pm0.01)}$ | $3.84_{(\pm0.01)}$ | $2.36_{(\pm0.01)}$ | $1.92_{(\pm0.00)}$ | $2.12_{(\pm0.00)}$ |
| | CD [51] | $4.52_{(\pm0.03)}$ | $3.55_{(\pm0.01)}$ | $3.98_{(\pm0.01)}$ | $3.59_{(\pm0.01)}$ | $4.00_{(\pm0.00)}$ | $3.79_{(\pm0.00)}$ |
| | AdaCAD [52] | $4.57_{(\pm0.05)}$ | $3.70_{(\pm0.10)}$ | $4.09_{(\pm0.08)}$ | $3.71_{(\pm0.02)}$ | $4.35_{(\pm0.01)}$ | $4.01_{(\pm0.01)}$ |
| | Entropy [53] | $4.56_{(\pm0.05)}$ | $4.14_{(\pm0.01)}$ | $4.34_{(\pm0.02)}$ | $3.81_{(\pm0.01)}$ | $4.39_{(\pm0.00)}$ | $4.08_{(\pm0.00)}$ |
| | CAD [26] | $4.52_{(\pm0.03)}$ | $3.55_{(\pm0.01)}$ | $3.98_{(\pm0.01)}$ | $3.77_{(\pm0.03)}$ | $4.43_{(\pm0.02)}$ | $4.08_{(\pm0.02)}$ |
| | COIECD [54] | $4.64_{(\pm0.10)}$ | $4.08_{(\pm0.02)}$ | $4.33_{(\pm0.01)}$ | $4.07_{(\pm0.00)}$ | $4.54_{(\pm0.00)}$ | $4.30_{(\pm0.00)}$ |
| | AGLA [57] | $4.80_{(\pm0.05)}$ | $4.28_{(\pm0.09)}$ | $4.52_{(\pm0.05)}$ | $3.74_{(\pm0.01)}$ | $4.10_{(\pm0.01)}$ | $3.91_{(\pm0.01)}$ |
| | VCD [58] | $4.70_{(\pm0.01)}$ | $4.14_{(\pm0.01)}$ | $4.40_{(\pm0.00)}$ | $3.62_{(\pm0.01)}$ | $4.12_{(\pm0.00)}$ | $3.85_{(\pm0.00)}$ |
| | **ALFAR (ours)** | $\mathbf{5.98}_{(\pm0.00)}$ | $\mathbf{5.67}_{(\pm0.00)}$ | $\mathbf{5.82}_{(\pm0.00)}$ | $\mathbf{4.55}_{(\pm0.00)}$ | $\mathbf{5.68}_{(\pm0.00)}$ | $\mathbf{5.05}_{(\pm0.00)}$ |
| Shikra | Regular [40] | $6.71_{(\pm0.03)}$ | $6.31_{(\pm0.01)}$ | $6.50_{(\pm0.02)}$ | $11.93_{(\pm0.01)}$ | $11.78_{(\pm0.01)}$ | $11.85_{(\pm0.01)}$ |
| | Parametric [40] | $5.76_{(\pm0.10)}$ | $6.10_{(\pm0.07)}$ | $5.92_{(\pm0.05)}$ | $7.61_{(\pm0.01)}$ | $6.25_{(\pm0.01)}$ | $6.86_{(\pm0.01)}$ |
| | CD [51] | $8.21_{(\pm0.00)}$ | $7.15_{(\pm0.01)}$ | $7.64_{(\pm0.01)}$ | $12.41_{(\pm0.00)}$ | $11.89_{(\pm0.00)}$ | $12.14_{(\pm0.00)}$ |
| | AdaCAD [52] | $8.30_{(\pm0.00)}$ | $7.11_{(\pm0.00)}$ | $7.66_{(\pm0.00)}$ | $12.87_{(\pm0.03)}$ | $12.53_{(\pm0.02)}$ | $12.70_{(\pm0.02)}$ |
| | Entropy [53] | $8.32_{(\pm0.03)}$ | $7.73_{(\pm0.11)}$ | $8.01_{(\pm0.05)}$ | $13.78_{(\pm0.02)}$ | $13.33_{(\pm0.01)}$ | $13.55_{(\pm0.02)}$ |
| | CAD [26] | $8.16_{(\pm0.06)}$ | $7.16_{(\pm0.02)}$ | $7.62_{(\pm0.00)}$ | $12.99_{(\pm0.03)}$ | $12.51_{(\pm0.02)}$ | $12.75_{(\pm0.03)}$ |
| | COIECD [54] | $8.32_{(\pm0.02)}$ | $7.73_{(\pm0.08)}$ | $8.01_{(\pm0.03)}$ | $14.46_{(\pm0.02)}$ | $14.21_{(\pm0.03)}$ | $14.33_{(\pm0.02)}$ |
| | AGLA [57] | $8.24_{(\pm0.01)}$ | $7.56_{(\pm0.05)}$ | $7.88_{(\pm0.01)}$ | $14.29_{(\pm0.02)}$ | $13.91_{(\pm0.01)}$ | $14.08_{(\pm0.01)}$ |
| | VCD [58] | $8.13_{(\pm0.05)}$ | $7.41_{(\pm0.03)}$ | $7.75_{(\pm0.04)}$ | $13.71_{(\pm0.03)}$ | $13.81_{(\pm0.03)}$ | $13.76_{(\pm0.03)}$ |
| | **ALFAR (ours)** | $\mathbf{8.61}_{(\pm0.01)}$ | $\mathbf{8.04}_{(\pm0.01)}$ | $\mathbf{8.31}_{(\pm0.01)}$ | $\mathbf{15.25}_{(\pm0.00)}$ | $\mathbf{15.11}_{(\pm0.01)}$ | $\mathbf{15.18}_{(\pm0.01)}$ |
| MiniGPT4 | Regular [16] | $4.38_{(\pm0.07)}$ | $3.00_{(\pm0.02)}$ | $3.56_{(\pm0.02)}$ | $12.69_{(\pm0.02)}$ | $12.38_{(\pm0.02)}$ | $12.53_{(\pm0.02)}$ |
| | Parametric [16] | $2.34_{(\pm0.01)}$ | $2.10_{(\pm0.01)}$ | $2.21_{(\pm0.00)}$ | $4.72_{(\pm0.01)}$ | $3.93_{(\pm0.01)}$ | $4.29_{(\pm0.01)}$ |
| | CD [51] | $4.28_{(\pm0.00)}$ | $2.59_{(\pm0.00)}$ | $3.22_{(\pm0.00)}$ | $14.49_{(\pm0.01)}$ | $14.43_{(\pm0.00)}$ | $14.46_{(\pm0.01)}$ |
| | AdaCAD [52] | $4.78_{(\pm0.00)}$ | $3.43_{(\pm0.01)}$ | $3.99_{(\pm0.00)}$ | $14.82_{(\pm0.02)}$ | $14.96_{(\pm0.02)}$ | $14.89_{(\pm0.02)}$ |
| | Entropy [53] | $4.80_{(\pm0.07)}$ | $2.91_{(\pm0.00)}$ | $3.62_{(\pm0.00)}$ | $14.66_{(\pm0.02)}$ | $14.66_{(\pm0.01)}$ | $14.66_{(\pm0.02)}$ |
| | CAD [26] | $4.97_{(\pm0.01)}$ | $3.44_{(\pm0.01)}$ | $4.07_{(\pm0.01)}$ | $14.83_{(\pm0.01)}$ | $14.94_{(\pm0.00)}$ | $14.88_{(\pm0.01)}$ |
| | COIECD [54] | $4.57_{(\pm0.03)}$ | $3.40_{(\pm0.01)}$ | $3.90_{(\pm0.00)}$ | $14.87_{(\pm0.01)}$ | $14.67_{(\pm0.02)}$ | $14.77_{(\pm0.02)}$ |
| | AGLA [57] | $4.67_{(\pm0.02)}$ | $3.63_{(\pm0.02)}$ | $4.09_{(\pm0.01)}$ | $14.31_{(\pm0.02)}$ | $13.92_{(\pm0.01)}$ | $14.11_{(\pm0.01)}$ |
| | VCD [58] | $4.52_{(\pm0.03)}$ | $3.43_{(\pm0.01)}$ | $3.90_{(\pm0.01)}$ | $14.46_{(\pm0.01)}$ | $14.30_{(\pm0.02)}$ | $14.38_{(\pm0.02)}$ |
| | **ALFAR (ours)** | $\mathbf{5.05}_{(\pm0.02)}$ | $\mathbf{3.87}_{(\pm0.01)}$ | $\mathbf{4.38}_{(\pm0.00)}$ | $\mathbf{15.16}_{(\pm0.01)}$ | $\mathbf{15.27}_{(\pm0.01)}$ | $\mathbf{15.05}_{(\pm0.01)}$ |

Similarly, we represent contextual knowledge by using the output logits that have context as additional inputs and perform attention reallocation to better utilize the context:

$$\text{logit}_\text{c} = \hat{\mathcal{M}}_\theta(y_t | q, I, c, y_{<t}) \tag{9}$$

where $\hat{\mathcal{M}}_\theta$ is the MLLM with attention reallocation. The reliability of the parametric and contextual knowledge can thus be measured by the attention weights that are assigned to the image and context tokens capturing the correlation among tokens [59, 60, 61]:

$$\lambda_v^t = \sum_{i \in S_I} \mathbf{A}_{ti} \,, \quad \lambda_c^t = \sum_{j \in S_c} \mathbf{A}_{tj} \tag{10}$$

Table 3: VQA Accuracy comparison on discriminative multi-choice VQA datasets over three runs.

| Model | Decoding | InfoSeek | ViQuAE | Model | Decoding | InfoSeek | ViQuAE |
|---|---|---|---|---|---|---|---|
| LLaVA-1.5 | Regular [14] | $51.97_{(\pm0.42)}$ | $53.32_{(\pm0.20)}$ | InstructBLIP | Regular [10] | $23.44_{(\pm0.89)}$ | $19.82_{(\pm0.12)}$ |
| | Parametric [14] | $39.15_{(\pm0.02)}$ | $51.06_{(\pm0.16)}$ | | Parametric [10] | $8.73_{(\pm0.23)}$ | $6.53_{(\pm0.35)}$ |
| | CD [51] | $49.95_{(\pm0.20)}$ | $52.56_{(\pm0.03)}$ | | CD [51] | $22.95_{(\pm0.63)}$ | $21.76_{(\pm0.51)}$ |
| | CAD [26] | $52.08_{(\pm0.16)}$ | $52.99_{(\pm0.23)}$ | | CAD [26] | $26.56_{(\pm0.56)}$ | $23.18_{(\pm0.30)}$ |
| | AdaCAD [52] | $52.30_{(\pm0.04)}$ | $52.99_{(\pm0.30)}$ | | AdaCAD [52] | $27.07_{(\pm0.05)}$ | $23.64_{(\pm0.08)}$ |
| | Entropy [53] | $53.33_{(\pm0.07)}$ | $54.26_{(\pm0.05)}$ | | Entropy [53] | $27.50_{(\pm0.07)}$ | $23.19_{(\pm0.39)}$ |
| | COIECD [54] | $52.08_{(\pm0.21)}$ | $52.99_{(\pm0.23)}$ | | COIECD [54] | $25.65_{(\pm0.05)}$ | $20.84_{(\pm0.05)}$ |
| | VCD [58] | $53.87_{(\pm0.07)}$ | $55.13_{(\pm0.09)}$ | | VCD [58] | $23.31_{(\pm0.07)}$ | $20.28_{(\pm0.53)}$ |
| | AGLA [57] | $53.53_{(\pm0.50)}$ | $54.24_{(\pm0.21)}$ | | AGLA [57] | $21.24_{(\pm0.55)}$ | $16.77_{(\pm0.30)}$ |
| | **ALFAR (ours)** | $\mathbf{58.35}_{(\pm0.21)}$ | $\mathbf{55.91}_{(\pm0.13)}$ | | **ALFAR (ours)** | $\mathbf{35.65}_{(\pm0.09)}$ | $\mathbf{24.11}_{(\pm0.07)}$ |
| Shikra | Regular [40] | $19.41_{(\pm0.12)}$ | $17.73_{(\pm0.01)}$ | MiniGPT-4 | Regular [16] | $25.83_{(\pm1.42)}$ | $24.06_{(\pm0.46)}$ |
| | Parametric [40] | $9.65_{(\pm0.14)}$ | $10.90_{(\pm0.04)}$ | | Parametric [16] | $19.73_{(\pm0.57)}$ | $20.42_{(\pm1.22)}$ |
| | CD [51] | $24.83_{(\pm0.20)}$ | $21.38_{(\pm0.03)}$ | | CD [51] | $26.55_{(\pm0.31)}$ | $20.46_{(\pm0.76)}$ |
| | CAD [26] | $24.51_{(\pm0.06)}$ | $21.68_{(\pm0.15)}$ | | CAD [26] | $27.58_{(\pm0.43)}$ | $23.39_{(\pm0.07)}$ |
| | AdaCAD [52] | $23.95_{(\pm0.18)}$ | $21.51_{(\pm0.01)}$ | | AdaCAD [52] | $28.01_{(\pm0.17)}$ | $23.05_{(\pm0.61)}$ |
| | Entropy [53] | $24.12_{(\pm0.03)}$ | $21.99_{(\pm0.08)}$ | | Entropy [53] | $28.84_{(\pm0.68)}$ | $22.59_{(\pm0.07)}$ |
| | COIECD [54] | $24.18_{(\pm0.13)}$ | $21.68_{(\pm0.15)}$ | | COIECD [54] | $29.44_{(\pm0.01)}$ | $25.94_{(\pm0.18)}$ |
| | VCD [58] | $25.76_{(\pm0.14)}$ | $23.11_{(\pm0.72)}$ | | VCD [58] | $28.74_{(\pm0.10)}$ | $25.45_{(\pm0.05)}$ |
| | AGLA [57] | $26.26_{(\pm0.29)}$ | $22.72_{(\pm0.15)}$ | | AGLA [57] | $29.28_{(\pm0.87)}$ | $27.87_{(\pm0.08)}$ |
| | **ALFAR (ours)** | $\mathbf{28.11}_{(\pm0.01)}$ | $\mathbf{23.42}_{(\pm0.29)}$ | | **ALFAR (ours)** | $\mathbf{30.88}_{(\pm0.12)}$ | $\mathbf{32.28}_{(\pm0.02)}$ |
| LLaVA-Next | Regular [62] | $53.90_{(\pm0.26)}$ | $55.44_{(\pm0.12)}$ | Qwen2.5-VL | Regular [63] | $53.00_{(\pm0.18)}$ | $53.84_{(\pm0.01)}$ |
| | Parametric [62] | $42.50_{(\pm0.12)}$ | $53.94_{(\pm0.11)}$ | | Parametric [63] | $45.96_{(\pm0.32)}$ | $53.28_{(\pm0.15)}$ |
| | CD [51] | $51.43_{(\pm0.23)}$ | $51.38_{(\pm0.01)}$ | | CD [51] | $56.23_{(\pm0.23)}$ | $59.33_{(\pm0.15)}$ |
| | CAD [26] | $53.83_{(\pm0.13)}$ | $52.84_{(\pm0.27)}$ | | CAD [26] | $58.67_{(\pm0.25)}$ | $60.92_{(\pm0.20)}$ |
| | AdaCAD [52] | $55.60_{(\pm0.34)}$ | $54.87_{(\pm0.13)}$ | | AdaCAD [52] | $57.87_{(\pm0.05)}$ | $60.89_{(\pm0.28)}$ |
| | Entropy [53] | $54.43_{(\pm0.20)}$ | $55.60_{(\pm0.15)}$ | | Entropy [53] | $58.67_{(\pm0.13)}$ | $60.66_{(\pm0.19)}$ |
| | COIECD [54] | $53.93_{(\pm0.11)}$ | $54.17_{(\pm0.33)}$ | | COIECD [54] | $59.23_{(\pm0.10)}$ | $60.59_{(\pm0.02)}$ |
| | VCD [58] | $53.27_{(\pm0.27)}$ | $54.32_{(\pm0.03)}$ | | VCD [58] | $58.56_{(\pm0.32)}$ | $60.12_{(\pm0.33)}$ |
| | AGLA [57] | $54.03_{(\pm0.25)}$ | $54.65_{(\pm0.34)}$ | | AGLA [57] | $59.13_{(\pm0.25)}$ | $61.08_{(\pm0.13)}$ |
| | **ALFAR (ours)** | $\mathbf{58.47}_{(\pm0.21)}$ | $\mathbf{59.29}_{(\pm0.21)}$ | | **ALFAR (ours)** | $\mathbf{61.57}_{(\pm0.18)}$ | $\mathbf{63.22}_{(\pm0.08)}$ |

Finally, based on the adaptive weights, the two types of knowledge are fused dynamically at each decoding step $t$:

$$p(y_t) \sim \mathrm{softmax}\left[(1 + \frac{\lambda_c^t}{\lambda_v^t})\,\mathrm{logit_c} - (1 - \frac{\lambda_v^t}{\lambda_c^t})\,\mathrm{logit_p}\right] \tag{11}$$

# 5 Experiment

## 5.1 Experimental Settings

**Datasets.** We conduct experiments over three types of knowledge-intensive datasets: (1) Free-form generative datasets including **Human** [23], a high-quality info-seeking dataset curated and verified by experts, and INFOSEEK$_{wiki}$ [23] which encompasses diverse entities from Wikidata. For INFOSEEK$_{wiki}$, we adopt its **Validation** set for evaluations to be aligned with prior studies [27, 28]. (2) Multi-choice discriminative datasets including **Infoseek** [23, 32] and **ViQuAE** [24, 32] which are both multi-choice knowledge-intensive datasets that are collected for assessing cross-modality knowledge conflicts as described in [32]. (3) Knowledge-based datasets including **OK-VQA** [64], **AOK-VQA** [65] and **Encyclopedic VQA** (E-VQA) [31], which are widely adopted for evaluations of tasks that require commonsense knowledge. Additional details about the datasets and knowledge bases are listed in the Appendix A5.

**MLLM baselines and SOTA methods.** We perform evaluations by using four representative MLLMs as backbones: LLaVA-1.5 (7B and 13B) [14], InstructBLIP (7B and 13B) [10], Shikra (7B) [40], MiniGPT-4 (7B) [16], LLaVA-Next (7B) [62], and Qwen2.5-VL (3B) [63]. For benchmarking, we select several SOTA training-free decoding methods that aim to mitigate knowledge conflicts in LLMs: Contrastive Decoding (CD) [51], Adaptive Context-Aware Decoding (AdaCAD) [52], Entropy-based decoding (Entropy) [53], Context-Aware Decoding (CAD) [26] and COntextual Information-Entropy Constraint Decoding (COIECD) [54]. In addition, we also benchmark with two representative

hallucination mitigation methods, including Visual Contrastive Decoding (VCD) [58] and Assembly of Global and Local Attention (AGLA) [57].

**Implementation details.** For knowledge retrieval, we employ the vision encoder of CLIP-ViT-L/14-336 [66] as the retriever and append the first retrieved knowledge to the prompt as context. The scaled factor $k$ is set to 0.4 to avoid excessive adjustment. For the knowledge base, we use Wikipedia dumps provided by [23] and select items with associated images for retrieval. Multinomial sampling serves as the decoding strategy. We denote MLLM inference with retrieved knowledge as *Regular* and without retrieved knowledge as *Parametric*. We follow prior studies [26, 53, 58] and adopt adaptive plausibility constraints [51] for fair comparisons. All experiments are conducted on four NVIDIA RTX 3090 GPUs. All compared methods are reproduced by us according to their released codes or original papers.

## 5.2 Experimental Results

**Experiments on free-form datasets.** Tab. 2 shows experimental results of four representative MLLMs [14, 10, 40, 16] over two free-form generative knowledge-intensive datasets [23]. We can see that the proposed ALFAR consistently outperforms the *Regular* decoding strategy by substantial margins (averaged around 2.5% in overall accuracy) across all MLLMs and datasets. Additionally, ALFAR surpasses state-of-the-art decoding methods as well, demonstrating its effectiveness in the better utilization of contextual knowledge.

**Experiments on multi-choice datasets.** Tab. 3 presents experimental results of six MLLMs [14, 10, 40, 16, 62, 63] over two multi-choice discriminative datasets [32, 24]. Notably, ALFAR achieves an average improvement of 6.6% over Regular decoding and consistently surpasses state-of-the-art decoding strategies by substantial margins, underscoring its effectiveness in diverse tasks. Moreover, we observe that LLaVA-1.5 [14] demonstrates a stronger instruction-following capability compared to other models, enabling it to produce more correctly formatted outputs.

**Experiments on knowledge-based datasets.** In addition to entity knowledge-based datasets [32, 24], we conduct experiments on commonsense knowledge-based datasets, OK-VQA [64], AOK-VQA [65] and Encyclopedic VQA (E-VQA) [31] with LLaVA-1.5 [14]. As shown in Tab. 4, ALFAR surpasses Regular decoding by 15.2% and consistently outperforms state-of-the-art decoding strategies, underscoring its effectiveness in addressing a broader range of knowledge-intensive tasks.

Table 4: VQA Accuracy comparison on the knowledge-based VQA datasets with LLaVA-1.5 [14] over three runs.

| Model | OK-VQA | AOK-VQA | E-VQA |
|---|---|---|---|
| Regular [14] | $46.17_{(\pm 0.12)}$ | $44.13_{(\pm 0.00)}$ | $19.14_{(\pm 2.83)}$ |
| Parametric [14] | $45.13_{(\pm 0.40)}$ | $43.23_{(\pm 1.02)}$ | $5.34_{(\pm 0.01)}$ |
| CD [51] | $55.00_{(\pm 0.03)}$ | $51.67_{(\pm 0.44)}$ | $28.62_{(\pm 0.68)}$ |
| CAD [26] | $56.43_{(\pm 0.46)}$ | $53.93_{(\pm 0.56)}$ | $28.62_{(\pm 0.76)}$ |
| AdaCAD [52] | $57.10_{(\pm 0.04)}$ | $54.40_{(\pm 0.48)}$ | $28.33_{(\pm 0.42)}$ |
| Entropy [53] | $56.27_{(\pm 0.05)}$ | $53.93_{(\pm 0.26)}$ | $29.24_{(\pm 0.11)}$ |
| COIECD [54] | $56.43_{(\pm 0.46)}$ | $53.93_{(\pm 0.56)}$ | $28.24_{(\pm 0.74)}$ |
| VCD [58] | $57.80_{(\pm 0.13)}$ | $57.40_{(\pm 0.12)}$ | $27.71_{(\pm 0.33)}$ |
| AGLA [57] | $57.53_{(\pm 0.05)}$ | $55.40_{(\pm 0.63)}$ | $28.29_{(\pm 0.56)}$ |
| **ALFAR (ours)** | $\mathbf{60.83}_{(\pm 0.01)}$ | $\mathbf{59.93}_{(\pm 0.02)}$ | $\mathbf{29.57}_{(\pm 0.08)}$ |

## 6 Discussion

### 6.1 Ablation study

We conduct ablation studies on both multi-choice and commonsense knowledge-based datasets [32, 65] to assess the effectiveness of each design in the proposed ALFAR model with LLaVA-1.5 [14]. As shown in Tab. 5, the **Attention Reallocation** enables MLLMs to better utilize retrieved knowledge, thereby enhancing overall performance. The **Logits Fusion** mitigates knowledge conflicts, allowing MLLMs to integrate retrieved knowledge

Table 5: Experimental results of ablation study with different model variants.

| Variants | InfoSeek | AOK-VQA |
|---|---|---|
| Regular [14] | 51.97 | 44.13 |
| + Attention Reallocation | 53.42 | 46.30 |
| + Logits Fusion | 55.83 | 55.90 |
| + Adaptive Weights | 58.35 | 59.93 |

and improve overall performance effectively. Moreover, applying **Adaptive Weights** during logits fusion helps MLLMs better leverage both parametric and contextual knowledge while reducing the impact of noise from the retrieved context, further contributing to performance gains.

## 6.2 Effect of different retrievers

We investigate the impact of different retrievers and retrieval strategies on recall and model performance on the InfoSeek dataset [32]. As shown in Tab. 6, our model consistently enhances performance across all configurations, demonstrating its robustness to retrieval noise. Additionally, retrieving knowledge based on the input query yields low recall due to its limited information about entities in the image. Despite this limitation, our model achieves consistent performance improvements even under low retrieval recall, owing to its adaptive fusion strategy that balances parametric and contextual knowledge.

Table 6: Experimental results with different CLIP retrievers. '-T' means using the input query for knowledge retrieval.

| Retriever | Recall | LLaVA | Ours |
|---|---|---|---|
| CLIP-B/16 [66] | 38.27 | 45.93 | **51.60** |
| CLIP-L/14 [66] | 56.53 | 51.23 | **57.60** |
| CLIP-L/14-336 [66] | 58.37 | 51.97 | **58.35** |
| CLIP-B/16-T [66] | 5.56 | 32.97 | **35.43** |
| CLIP-L/14-T [66] | 6.00 | 33.40 | **36.23** |
| CLIP-L/14-336-T [66] | 5.73 | 33.37 | **36.63** |

## 6.3 Inference Efficiency

In this section, we conduct a detailed analysis of the inference efficiency of the proposed model in comparison with representative baseline methods. Specifically, we examine how the incorporation of parametric knowledge modeling affects the inference time under both discriminative and generative tasks. Compared with the baseline MRAG, our model introduces only a modest computational overhead, which mainly arises from the additional operations required to encode and integrate parametric knowledge. Nevertheless, this extra cost remains limited, as the parametric knowledge component without contextual input is relatively concise. Moreover, all compared methods except the baseline also require modeling of both contextual

Table 7: Inference time of different methods over one sample with LLaVA-1.5 [14].

| Variants | InfoSeek | Human |
|---|---|---|
| Regular [14] | 0.46s (1.00x) | 0.50s (1.00x) |
| CD [51] | 0.62s (1.35x) | 0.72s (1.44x) |
| CAD [26] | 0.62s (1.34x) | 0.72s (1.44x) |
| AdaCAD [52] | 0.62s (1.35x) | 0.73s (1.45x) |
| Entropy [53] | 0.63s (1.37x) | 0.73s (1.46x) |
| COIECD [54] | 0.62s (1.35x) | 0.73s (1.45x) |
| VCD [58] | 0.63s (1.38x) | 0.74s (1.47x) |
| AGLA [57] | 0.83s (1.81x) | 1.02s (2.04x) |
| **ALFAR (ours)** | 0.62s (1.35x) | 0.73s (1.46x) |

and parametric knowledge. Consequently, the overall inference cost of our model is comparable to that of these methods. We evaluate all models on both the discriminative multi-choice dataset Infoseek [32] and the free-form generative dataset Human [23] using a single input sample. The results summarized in Tab. 7 demonstrate that our method achieves similar inference times.

## 6.4 Qualitative Examples

Fig. 6 shows a qualitative comparison of four decoding approaches. It can be observed that Vanilla LLaVA [14] without context produces a false response due to the lack of knowledge about the entity in the image. LLaVA with MRAG [27] incorporates the contextual knowledge but still produces the same incorrect response as Vanilla LLaVA [14], primarily due to knowledge conflicts that cause MLLMs to favor their parametric knowledge. LLaVA with CAD [26] can mitigate knowledge conflicts effectively by reducing the influence of parametric knowledge. However, it produces a false response as well, mainly due to attention bias with excessive focus on images and uniform attention toward context. In contrast, ALFAR introduces attention reallocation and adaptive logits fusion, enabling MLLMs to prioritize query-relevant contextual knowledge and produce accurate responses.

## 6.5 ALFAR on MLLM Scalability

Tab. 8 presents experimental results of the 13B variants of LLaVA-1.5 [14] and InstructBLIP [10] over the InfoSeek dataset [32]. Notably, ALFAR consistently improves performance across both models, demonstrating its superior scalability with respect to the model size. Interestingly, we observe a performance decline in

Table 8: Experiments with larger MLLMs.

| Decoding | LLaVA-13B | InstructBLIP-13B |
|---|---|---|
| Parametric | 44.40 | 8.93 |
| Regular | 55.83 | 18.70 |
| **ALFAR (ours)** | **59.63** | **27.63** |

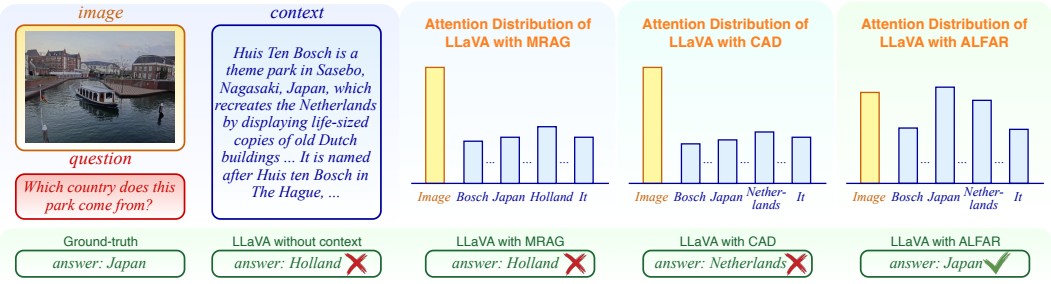

Figure 6: Illustration of generated responses and corresponding attention distributions across different decoding methods, using LLaVA-1.5 [14] as the backbone model.

InstructBLIP 13B compared to the 7B variant, which may be attributed to the model's increased reliance on parametric knowledge.

## 6.6 Effect of Different Decoding Strategies

Beyond the multinomial sampling strategy discussed in this paper, we further investigate the robustness of the proposed ALFAR framework under diverse decoding settings. To this end, we conduct experiments using LLaVA-1.5 [14] on the multi-choice InfoSeek dataset [32], and evaluate six additional decoding strategies commonly adopted in MLLM generation. These strategies include Top-P sampling [67] with $p = 0.7$, Top-K sampling [68] with $k = 50$, greedy decoding [69], temperature sampling [70] with $t = 0.5$, Top-P sampling with

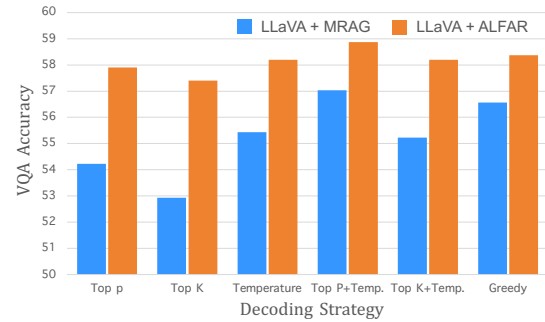

Figure 5: Experimental results with different decoding strategies for response generation.

temperature ($p = 0.7$, $t = 0.5$), and Top-K sampling with temperature ($k = 50$, $t = 0.5$). As illustrated in Fig. 5, ALFAR consistently enhances model performance across all decoding strategies. This consistent gain highlights the generalizability of ALFAR's learning principle and suggests that it effectively complements various decoding schemes. Moreover, the results indicate that ALFAR can serve as a robust and plug-and-play enhancement to existing MLLMs, ensuring stable performance regardless of decoding configuration.

## 7 Conclusion

In this paper, we examine representative MLLMs and find that they often struggle to fully utilize retrieved knowledge for knowledge-intensive tasks. We attribute this limitation to two key factors: *attention bias* toward different tokens and *knowledge conflicts* between parametric and contextual knowledge. To address these challenges, we introduce *Adaptive Logits Fusion and Attention Reallocation (ALFAR)*, a training-free and plug-and-play approach that enhances MLLM performance by dynamically reallocating attention and harmonizing parametric and contextual knowledge. Specifically, ALFAR mitigates attention bias by adaptively shifting focus from visual tokens to context tokens based on query-context relevance. Furthermore, it decouples and balances parametric and contextual knowledge at the output logits, effectively resolving conflicts. Experiments across multiple MLLMs and benchmarks show that ALFAR consistently surpasses state-of-the-art methods by substantial margins without requiring additional training or external tools, highlighting its versatility and effectiveness for various knowledge-intensive tasks.

## 8 Acknowledgments

This work was supported by the National Science and Technology Major Project (2022ZD0117102), National Natural Science Foundation of China (62177038, 62293551, 62277042, 62377038), Project of China Knowledge Centre for Engineering Science and Technology, "LENOVO-XJTU" Intelligent Industry Joint Laboratory Project, The Youth Al Talents Fund of the Chinese Association of Automation under Major Program (HBRC-JKYZD-2024-311).

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

## A1 Limitation and Future Work

Despite its strong performance in enhancing knowledge utilization for MLLMs through adaptive logits fusion and attention reallocation, our approach has certain limitations. First, our framework requires access to MLLM parameters, making it inapplicable to black-box API-based models such as GPT-4 [71]. Extending our framework to black-box MLLMs represents a promising direction for future research. Additionally, we observe that MLLMs struggle to effectively extract relevant information from long contexts. Addressing this limitation by improving MLLMs' ability to leverage extended contexts will be another focus of future work.

## A2 Broader Impacts

The proposed model for enhancing knowledge utilization in MLLMs carries significant broader impacts. First, by addressing the critical issue of MRAG, our method enhances the reliability and trustworthiness of MLLMs. This improvement is essential for deploying these models in sensitive and high-stakes applications such as autonomous driving, medical diagnostics, and surveillance systems. Second, the insights and methods introduced in this paper contribute to the broader field of MLLMs, particularly in understanding and improving the knowledge utilization mechanisms within MLLMs. This advancement can spur further research and innovation in integrating visual and textual data, leading to more robust and versatile AI models.

## A3 Conflict Rate and Performance Drop

To quantify the conflict between parametric and contextual knowledge and its impact on model performance, we introduce two metrics: *Conflict Rate* and *Performance Drop*.

*Conflict Rate* measures the proportion of instances where parametric and contextual knowledge provide different information, and *Performance Drop* quantifies the decline in model performance due to knowledge conflict. Since parametric knowledge is implicitly embedded in model parameters and is not directly observable, we approximate its correctness by evaluating the model's outputs. Specifically, if the model (without external context) produces the correct answer, we assume its parametric knowledge is correct; otherwise, it is considered incorrect. Given access to ground-truth contextual knowledge, the *Conflict Rate* can be defined as the error rate of parametric knowledge, i.e., the proportion of incorrect responses generated by the vanilla model without input context:

$$\texttt{Conflict Rate} = \texttt{Err}(\mathcal{M}_\theta(y|q, I), \hat{y}) \tag{A1}$$

where $\texttt{Err}$ is a function that calculates the error rate of the output, $\mathcal{M}_\theta(y|q, I)$ is the output with only images and questions as inputs, and $\hat{y}$ is the ground-truth answer.

When correct contextual knowledge is available, the ideal model should achieve 100% accuracy in the absence of knowledge conflicts. However, influenced by knowledge conflicts, the model cannot achieve 100% accuracy, then we can define *Performance Drop* as the error rate of outputs when both parametric and contextual knowledge are used:

$$\texttt{Performance Drop} = \texttt{Err}(\mathcal{M}_\theta(y|q, I, c), \hat{y}) \tag{A2}$$

where $\mathcal{M}_\theta(y|q, I, c)$ is the output with ground-truth context as additional inputs.

## A4 Retrieval Recall

To investigate the retrieval recall from different retrieval rankings, we present the recall with Ground-Truth knowledge and knowledge from various retrieval rankings on the multi-choice InfoSeek dataset [23, 32] in Tab. A1. The low recall negatively impacts performance on knowledge-intensive VQA tasks, highlighting the necessity of developing a more effective retriever.

## A5 Dataset

We present statistics of different datasets and the corresponding knowledge bases in Tab. A2. Specifically, for Validation [23] and InfoSeek [32], we follow previous works [27] and adopt a knowledge

Table A1: Retrieval recall with Ground-Truth knowledge (GT) and knowledge from different retrieval rankings on the multi-choice InfoSeek dataset [23, 32].

| Index | GT | 1 | 2 | 3 | 4 |
|---|---|---|---|---|---|
| Recall | 100 | 58.37 | 10.57 | 5.07 | 3.07 |

Table A2: Statistics of the datasets and details of the knowledge bases used.

| Dataset | # VQA pairs | Knowledge Base |
|---|---|---|
| Validation [23] | 73,620 | Wikipedia [23] |
| Human [23] | 8,931 | Wikipedia [23] |
| InfoSeek [32] | 3,000 | Wikipedia [23] |
| ViQuAE [32] | 3,000 | Wikipedia [23] |
| OK-VQA [64] | 5,046 | GPT-3.5 [21] |
| AOK-VQA [65] | 1,145 | GPT-3.5 [21] |
| E-VQA [31] | 700 | Encyclopedia [31] |

base containing 1.7K entities derived from the original Wikipedia knowledge base [23]. For Human [23] and ViQuAE [24], we use the original Wikipedia knowledge base [23], selecting 73.6K entities accompanied by images for knowledge retrieval. For OK-VQA [64] and AOK-VQA [65], we utilize the knowledge base provided by [21], which was generated using GPT-3.5 [72]. For E-VQA [31], we select templated questions with images from the iNaturalist dataset [73] and use the corresponding ground-truth knowledge for inference. All evaluations are conducted using the official scripts.

## A6 Adaptive Plausibility Constraints

We follow prior studies [26, 53, 58] and adopt adaptive plausibility constraints [51] for fair comparisons. Specifically, calibrating the entire output distribution may penalize valid outputs from the original distribution and promote implausible outputs from the modified distribution. To mitigate this issue, we selectively consider tokens with high original probabilities and truncate other tokens as follows:

$$\mathcal{V}_{\text{token}}\left(y_{<i}\right) = \left\{y_i \in \mathcal{V} : p_\theta\left(y_i\right) \geq \beta \max_w p_\theta\left(w\right)\right\}$$
$$p\left(y_i\right) = 0, \text{ if } y_i \notin \mathcal{V}_{\text{token}}\left(y_{<i}\right)$$

(A3)

where $\mathcal{V}_{\text{token}}$ is the set of selected tokens and $\mathcal{V}$ is the output vocabulary. We select $\beta = 0.7$ to retain only high-probability tokens.

## A7 Effect of Intervention Layers

We investigate the impact of attention reallocation at different MLLM layers on the InfoSeek dataset [23] using LLaVA-1.5 [14], as summarized in Tab. A4. The results show that reallocating attention in shallow layers (layers 1-16) enhances model performance by mitigating attention bias toward image tokens, thereby improving the extraction of low-level features [30]. In contrast, applying attention reallocation in middle layers (layers 17-24) yields smaller gains, as these layers primarily handle multimodal alignment and feature aggregation [74], where attention bias is less severe. Notably, reallocating attention in late layers (layers 25-32) leads to the most substantial performance gains, as these layers are responsible for reasoning and directly affect output generation [74]. Furthermore, leveraging attention reallocation in both shallow (layers 1–8) and deep (layer 32) layers yields the best performance.

Table A4: An ablation study with different layers for attention reallocation.

| Intervention Layers | VQA Accuracy |
|---|---|
| None | 56.70 |
| $[1 - 8]$ | 58.08 |
| $[9 - 16]$ | 58.17 |
| $[17 - 24]$ | 57.57 |
| $[25 - 32]$ | 58.10 |
| $[1 - 8] \cup [32]$ | **58.67** |

## A8 Effect of different numbers of knowledge

We examine the effect of varying the amount of knowledge provided to MLLMs on retrieval recall and model performance on the InfoSeek dataset [23]. As shown in Tab. A3, appending additional knowledge to the prompt improves retrieval recall but has limited impact on model performance, as MLLMs often struggle to effectively utilize information from lengthy input contexts [75]. Our model addresses this limitation by guiding MLLMs based on query-context relevance. However, the modest performance gains underscore the need for future research on enhancing MLLMs' ability to process and leverage extended contexts.

Table A3: Experimental results with different numbers of knowledge using LLaVA-1.5 [14].

| #Knowledge | Recall | LLaVA [14] | Ours |
|---|---|---|---|
| 1 | 58.37 | 51.97 | **58.35** |
| 2 | 68.93 | 49.90 | **58.70** |
| 3 | 74.00 | 50.13 | **58.57** |
| 4 | 76.77 | 50.23 | **58.20** |

