# OpenReview forum: "Boosting Knowledge Utilization in Multimodal Large Language Models via Adaptive Logits Fusion and Attention Reallocation"
_NeurIPS.cc/2025/Conference — NeurIPS 2025 oral_

### Official Review · Reviewer_hLen · 2025-06-04

**Clarity:** 2
**Significance:** 3
**Originality:** 3
**Rating:** 4
**Confidence:** 3

**Summary:**

This paper proposes a method (named ALFAR) to address two claimed limits of existing Multimodal Large Langeuage Models (MLLMs) 1. attention bias towards image tokens over text tokens and 2. knowledge conflicts between parametric and contextual knowledge.
To do this, ALFAR modifies and reallocates attentions on tokens of different modalities, adaptively learns logit representations for both parametric and contextual knowledge, and fuse two knowledge for answer prediction.

**Questions:**

Questions
1. How is parametric and contextual knowledge formally defined? Is parametric knowledge limited to the knowledge acuired during training, and contextual knowledge limited to external data? as claimed in line 27-30 for MRAG? Is this contradictory from Fig. 4, where both logits for parameteric and contextual knowledge are derived from external data obeservations?
2. Can Eq. 11 be explained with more details? What does each term represent?
3. In Fig. 6, which part of the framework makes the attention be allocated more to Japan for LLaVa with ALFAR? Why doesn't the method promote other words such as Holland or Netherlands like LLaVA with MRAG or with CAD? Is the issue then really the bias of image tokens over text tokens? It seems to be accuracy identify which text tokens to pay more attention? It is not convincing that the bias attention towards image tokens is a problem or the root cause of incorrect answers.

Other minor questions
1. Should the y-axis of Fig. 2 be Attention Values (Sum) or alike instead of Distribution? What is the distribution in Fig. 2?

**Ethical Concerns:**

["NO or VERY MINOR ethics concerns only"]

**Final Justification:**

The response from authors has addressed some of my concern. I therefore choose to adjust my rating.

**Limitations:**

No.
The authors claim to have a Limitation section but is missing from the paper.

**Quality:**

2

**Strengths And Weaknesses:**

The paper is fair in quality and calrity, good in originality and significance.
- Clarity: The paper clearly motivates the problem, describes the workflow and experiments, and explains the results, although some contents need more articulation such as the definitions of the parametric and contextual knowledge
- Quality: The paper correctly formulates the problem, coherently develops the approach, and conducts experiments to support its claims, although some decisions should be examined and inspected, and some parts should be further explained. Please see Questions.
- Originality: The paper proposes a novel approach to reallocate attentions and fuse knowledge, although similar ideas for attention reallocations have been introduced by earlier work.
- Significance: The paper targets an important problem, but more validations are needed to enhance the significance of the work.

---

> ### Author Rebuttal · Authors · 2025-07-30
>
> Thanks for the constructive comments and suggestions. Below please find our clarification regarding your concerns.
>
> ---
>
> **[Q1]** Definition of parametric and contextual knowledge.
>
> **[A1]** Your understanding of parametric and contextual knowledge is correct, and this distinction **aligns with our definition** and is widely adopted in the LLM literature [1] [2].
>
> Regarding Figure 4, we would like to clarify **a possible misunderstanding**: in the upper part of the figure, the logits represent parametric knowledge, which **does not have access to the context (depicted by the absence of blue squares)**. In contrast, the lower part illustrates contextual knowledge, where the context (blue squares) is visible. This setup is consistent with the definitions of the two types of knowledge as used throughout our paper.
>
> We will revise the figure caption and description in the paper to make this distinction clearer.
>
> ---
>
> **[Q2]** Explanation about Equation 11.
>
> **[A2]** Thank you for your question. We will add more details in the revised manuscript to clarify the intent and components of Equation 11. Specifically:
> 1. $logit_c$ and $logit_p$ represent the logits derived from contextual knowledge and parametric knowledge, respectively.
>
> 2. $\lambda_{v}^{t}$ and $\lambda_{c}^{t}$ denote the summed attention weights over image tokens and context tokens at decoding step $t$. These values reflect the relative importance of the image and context at that time step.
> We will revise the explanation in the revised paper to improve the clarity and interpretability of this equation.
>
> ---
>
> **[Q3]** Question about Figure 6.
>
> **[A3]** Your statement is correct and **aligns well with the key contribution of our paper**: accurately identifying which context tokens to attend to is essential, and this is precisely what our model is designed to achieve.
>
> As shown in Equations (6) and (7), we **compute the relevance between the question and each context token** and use it as a guiding signal for attention reallocation. Such a design allows the model to **assign higher attention weights to context tokens that are more semantically aligned with the question**.
>
> In the case illustrated in Figure 6, our model correctly focuses on the token "Japan", as it appears near the word "Park", which is also a keyword in the question. This localized relevance enables the model to ground the answer more accurately. In contrast, LLaVA with MRAG or CAD, which lacks such fine-grained context control, tends to over-rely on visual cues (e.g., the European-style building in the image), leading to incorrect predictions like "Holland" or "Netherlands".
>
> In summary, our approach enables fine-grained control over context-level attention with question-context similarity, which dynamically reweights attention and reduces interference from less relevant image features. This leads to better knowledge utilization and more accurate predictions, which is aligned with your observations.
> We will revise the text to better highlight this point and clarify this contribution.
>
> ---
>
> **[Q4]** Y-axis of Figure 2.
>
> **[A4]** Thank you for the insightful question.
> In this figure, we first sum the attention weights assigned to image tokens and context tokens separately, and then **normalize these two values so that their sum equals one**. The normalization allows us to reflect the relative proportion of attention allocated to each type of token, rather than their absolute magnitudes.
> We referred to the Y-axis of Figure 2 as a “distribution” because it represents a normalized split of attention across the two modalities. To avoid confusion, we will revise the y-axis label to a more precise term like 'Normalized Attention Weights', and clarify the computation process in the main text.
>
> ---
>
> **[Q5]** Missing Limitation section.
>
> **[A5]** We apologize for the confusion. Due to space constraints, **the Limitation section was placed in the Appendix**. In the revised version, we will move it into the main paper to ensure better visibility and completeness of the discussion.
>
> ---
>
> Reference
>
> [1] Knowledge Conflicts for LLMs: A Survey.
>
> [2] Mitigating Knowledge Conflicts in Language Model-Driven Question Answering.

---

> > ### Comment · Reviewer_hLen · 2025-07-31
> >
> > Your response addresses some of my concern. I therefore choose to adjust my rating.

---

> > > ### Author Response · Authors · 2025-08-03
> > >
> > > Dear Reviewer hLen:
> > >
> > > We sincerely thank you for your thoughtful reading of our rebuttal and for reconsidering your evaluation. We appreciate your constructive comments and will make sure to include the corresponding analysis in the revised manuscript.
> > >
> > > Submission14912 Authors

---

### Official Review · Reviewer_aGW7 · 2025-06-15

**Clarity:** 4
**Significance:** 3
**Originality:** 3
**Rating:** 5
**Confidence:** 3

**Summary:**

This paper addresses the limitations of Multimodal Large Language Models (MLLMs) in knowledge-intensive tasks, especially their inability to fully utilize retrieved contextual knowledge. The authors propose ALFAR, a training-free and plug-and-play method that mitigates attention bias and resolves conflicts between parametric and retrieved knowledge. Experiments show that ALFAR significantly improves LVLM performance across various datasets, surpassing existing state-of-the-art approaches.

**Questions:**

See Weaknesses.

**Ethical Concerns:**

["NO or VERY MINOR ethics concerns only"]

**Final Justification:**

Author has conducted additional experiments to validate applicability across a broader spectrum of vision-language tasks.

**Limitations:**

Yes.

**Paper Formatting Concerns:**

No.

**Quality:**

3

**Strengths And Weaknesses:**

Pros:

1. The paper offers novel insights by identifying attention bias and knowledge conflicts as critical obstacles to effective knowledge utilization in Multimodal Large Language Models (MLLMs).

2. It proposes ALFAR, a training-free and plug-and-play method that effectively reallocates attention and balances parametric and contextual knowledge.

3. Extensive experiments demonstrate ALFAR’s superior performance and wide applicability across diverse generative and discriminative multimodal tasks.

Cons:

1. While your method has shown strong performance on a range of VQA datasets, it would further strengthen the contribution to demonstrate its applicability across a broader spectrum of vision-language tasks. In particular, evaluating on image captioning tasks could highlight the versatility of your approach in generating context-aware and knowledge-grounded descriptions.

2. It would be helpful to cite a recent survey on Multimodal Large Language Models to provide readers with a comprehensive background and better contextualize your contribution within the broader MLLM research landscape [1].

[1] "MM-LLMs: Recent Advances in MultiModal Large Language Models." Findings of the Association for Computational Linguistics ACL 2024. 2024.

---

> ### Author Rebuttal · Authors · 2025-07-30
>
> Thank you for the constructive comments and suggestions. Below please find our clarification regarding your concerns.
>
> ---
>
> **[W1]** Applicability across a broader spectrum of vision-language tasks.
>
> **[R1]** We appreciate the reviewer’s valuable suggestion. As suggested, we conducted **additional experiments on two vision-language tasks using LLaVA-1.5 and InstructBLIP: image captioning and multimodal hallucination mitigation**.
>
> For image captioning, we selected 500 images from the COCO dataset and evaluated caption reliability using the CHAIR metric (lower is better) and caption detailedness using Recall (higher is better). For multimodal hallucination mitigation, we conducted experiments on the POPE dataset, with accuracy and F1 score measuring the model’s ability to reduce hallucinations.
>
> **Evaluation on image captioning**
> | Model              | $CHAIR_i$ ↓ | $CHAIR_s$ ↓ | Recall ↑ |
> |--------------------|---------------------|----------------------|----------|
> | LLaVA              | 51.0                | 15.2                 | 75.2     |
> | LLaVA + Ours       | 48.3                | 14.5                 | 78.3     |
> | InstructBLIP       | 54.0                | 18.1                 | 71.1     |
> | InstructBLIP + Ours| 50.2                | 13.7                 | 72.6     |
>
> **Evaluation on multimodal hallucination mitigation**
> | Model               | Accuracy | F1   |
> |---------------------|----------|------|
> | LLaVA               | 83.5     | 82.3 |
> | LLaVA + Ours        | 85.8     | 85.5 |
> | InstructBLIP        | 80.4     | 80.9 |
> | InstructBLIP + Ours | 83.2     | 83.1 |
>
> As shown in the tables, **our approach consistently improves performance across all metrics and tasks** by effectively retrieving similar images and relevant descriptions. These new experiments further highlight the versatility and robustness of our method in enhancing vision-language models to generate more reliable and detailed outputs.
>
> ---
>
> **[W2]** Missing references.
>
> **[R2]** We appreciate the reviewer’s valuable suggestion. We will **include the recommended paper along with other recent relevant works** to ensure a comprehensive and up-to-date review of the MLLM research landscape in the revised manuscript.

---

> > ### Comment · Reviewer_aGW7 · 2025-08-04
> >
> > Thanks for your response. I have updated my rating.

---

> > > ### Author Response · Authors · 2025-08-04
> > >
> > > Dear Reviewer aGW7,
> > >
> > > We sincerely thank you for reading our rebuttal and for updating your rating. We greatly appreciate your constructive suggestions and will ensure that the corresponding analyses, experiments, and references are incorporated into the revised manuscript.
> > >
> > > Sincerely,
> > > Authors of Submission 14912

---

### Official Review · Reviewer_4mPJ · 2025-06-23

**Clarity:** 3
**Significance:** 3
**Originality:** 3
**Rating:** 5
**Confidence:** 4

**Summary:**

This paper addresses a key limitation of multimodal large language models (MLLMs): their inability to effectively leverage retrieved external knowledge. To overcome this, the authors propose ALFAR, a training-free and plug-and-play method designed to mitigate attention bias and resolve knowledge conflicts in MLLMs when tackling knowledge-intensive tasks. Through its two core components—Attention Reallocation and Adaptive Logits Fusion—ALFAR significantly enhances knowledge utilization performance. Experiments conducted on four MLLMs across six knowledge-centric datasets demonstrate that ALFAR consistently and substantially outperforms prior state-of-the-art decoding methods.

**Questions:**

1. The analysis of different intervention layers is interesting but might be better placed in the main paper rather than as supplementary material.
2. The definitions of certain variables (e.g., β, γ, λ) should be clarified and explained more thoroughly.

**Ethical Concerns:**

["NO or VERY MINOR ethics concerns only"]

**Final Justification:**

The author's response effectively addressed my concerns, and I believe this article is worth accepting. Therefore, I decided to improve my rating.

**Limitations:**

Yes

**Quality:**

3

**Strengths And Weaknesses:**

# Strength
1. The authors accurately identify two critical issues that limit knowledge utilization in MLLMs, attention bias and knowledge conflicts, and provide detailed analyses and validation of their causes and impacts.
2. The proposed two modules, attention reallocation and adaptive logits fusion, offer a novel and effective perspective to enhance MLLM performance in knowledge-intensive tasks.
3. Experiments across multiple representative MLLMs and diverse datasets validate the effectiveness of the proposed framework.
4. As a training-free and plug-and-play module, ALFAR can be directly applied to many MLLMs in MRAG pipelines.

# Weakness
1. The evaluated LVLMs are somewhat outdated; the framework’s effectiveness should be validated on newer LVLMs (e.g., Qwen2.5-vl, LLaVA-Next).
2. Additional analysis and validation of the framework’s robustness to varying retrieval quality would be beneficial.
3. The details of the compared methods are not clearly described.

---

> ### Author Rebuttal · Authors · 2025-07-30
>
> Thank you for the constructive comments and suggestions. Below please find our clarification regarding your concerns.
>
> ---
>
> **[W1]** Validation on newer LVLMs.
>
> **[R1]** We appreciate the reviewer’s suggestion. In response, we have conducted **additional experiments on two newer LVLMs, LLaVA-Next and Qwen2.5-VL, using the Infoseek dataset. The tabulated results below show that our method improves with both models consistently, validating its effectiveness and robustness on more recent architectures.
>
> | Method   | LLaVA-Next | Qwen2.5-VL |
> |----------|------------|-------------|
> | Parameter | 45.96     | 42.50       |
> | Regular   | 53.00     | 53.90       |
> | CD        | 56.23     | 51.43       |
> | CAD       | 58.67     | 53.83       |
> | ADCAD     | 57.87     | 55.60       |
> | Entropy   | 58.67     | 54.43       |
> | COIECD    | 59.23     | 53.93       |
> | VCD       | 58.56     | 53.27       |
> | AGLA      | 59.13     | 54.03       |
> | Ours      | 61.57     | 58.47       |
>
> ---
>
> **[W2]** Framework’s robustness.
>
> **[R2]** We appreciate the reviewer’s suggestion. We would share that the robustness of our framework has been validated from four perspectives:
>
> 1. **Robustness to knowledge quality**: As shown in Figure 1 of the submitted paper, our model achieves consistent improvements when the quality of retrieved knowledge varies. This indicates that the model can adaptively determine which knowledge to rely on, reflecting its robustness.
>
> 2. **Robustness across MLLMs**: As shown in Tables 2, 3, and 4, our model integrates effectively with four different MLLMs, demonstrating its compatibility and generalizability across diverse architectures.
>
> 3. **Robustness to retrievers**: As shown in Table 6, our model consistently improves MRAG performance across different retrievers, highlighting its tolerance to retrieval noise.
>
> 4. **Robustness to model scale**: As shown in Table 7, our method yields consistent gains on 13B-scale MLLMs, confirming its scalability with respect to model size.
>
> ---
>
> **[W3]** Details of the compared methods.
>
> **[R3]** We appreciate the reviewer’s suggestion. For all compared methods, we follow the implementation and model settings as described in their original papers to ensure a fair comparison. We will include more detailed descriptions of these methods and their configurations in the revised version for better clarity and reproducibility.
>
> ---
>
> **[Q1]** Analysis of different intervention layers might be better placed in the main paper.
>
> **[A1]** Thank you for the helpful suggestion. We agree that the analysis of different intervention layers is an important component of our study. Accordingly, we will revise the paper to include this experiment in the main paper as suggested, ensuring that readers can better appreciate its contribution.
>
> ---
>
> **[Q2]** Definitions of certain variables.
>
> **[A2]** We apologize for the confusion. The definitions of the variables are as follows:
>
> 1. As described in Line 176, $\beta$ is a factor that controls the strength of adjustment for image tokens.
>
> 2. As described in Line 190, $\gamma$ controls the strength of adjustment for each context token.
>
> 3. As defined in Equation 10, $\lambda$ represents the summed attention weights over image and context tokens, used for weighted fusion.
>
> We ensure that all these definitions will be stated clearly and prominently in the revised paper to improve readability and understanding.

---

> > ### Comment · Reviewer_4mPJ · 2025-08-04
> >
> > The author's response effectively addressed my concerns, and I believe this article is worth accepting. Therefore, I decided to improve my rating.

---

> > > ### Author Response · Authors · 2025-08-04
> > >
> > > Dear Reviewer 4mPJ,
> > >
> > > We sincerely thank you for reading our rebuttal and for recognizing the value of our work. We greatly appreciate your constructive feedback and will ensure that the corresponding analyses, experiments, and clarifications of definitions are incorporated into the revised manuscript.
> > >
> > > Authors of Submission 14912

---

### Official Review · Reviewer_NFNU · 2025-07-03

**Clarity:** 3
**Significance:** 3
**Originality:** 2
**Rating:** 5
**Confidence:** 3

**Summary:**

This work introduces a training-free, plug-and-play framework that tackles attention bias and knowledge conflicts in multimodal large language models through adaptive attention reallocation and logits fusion, yielding consistent performance gains of 6–15% across multiple VQA benchmarks and backbone models without additional training.

**Questions:**

My main concerns are in the method section, please see the weakness above.

**Ethical Concerns:**

["NO or VERY MINOR ethics concerns only"]

**Final Justification:**

Strengths:

1. The method is simple and effective with training-free and plug-and-play manners, requiring no additional model training or external tools for integration.

2. The preliminary findings on attention bias and knowledge conflicts, which hinder the effective utilization of retrieved context, are convincing.

3. Extensive evaluation across multiple MLLMs and diverse VQA benchmarks demonstrates consistent performance gains (6–15% improvements). Ablation studies and tests with different retrievers confirm the effectiveness and robustness of each component and the overall approach.

After rebuttal: I was very impressed and satisfied with how authors addressed my concerns regarding experiments on inference costs and comparison with other existing approaches. Thus I decided to raise my score.

**Limitations:**

The limitation discussions are sound and clear to me.

**Quality:**

3

**Strengths And Weaknesses:**

Strengths:

1. The method is simple and effective with training-free and plug-and-play manners, requiring no additional model training or external tools for integration.

2. The preliminary findings on attention bias and knowledge conflicts, which hinder the effective utilization of retrieved context, are convincing.

3. Extensive evaluation across multiple MLLMs and diverse VQA benchmarks demonstrates consistent performance gains (6–15% improvements). Ablation studies and tests with different retrievers confirm the effectiveness and robustness of each component and the overall approach.

Weaknesses:

1. The proposed method appears incremental, as the techniques involving attention weighting and logits fusion are already well-known and mainly borrowed from previous works.

2. A comparative inference cost between the proposed method and the existing approach (MRAG) is not provided.

---

> ### Author Rebuttal · Authors · 2025-07-30
>
> Thanks for the constructive comments and suggestions. Below please find our clarification regarding your concerns.
>
> ---
>
> **[W1]** Attention weighting and logits fusion are borrowed from previous works.
>
> **[R1]** We acknowledge that attention weighting and logits fusion have been explored in prior studies, represented by [1] and [2]. However, our approach differs significantly in motivation, method design, and empirical performance, as outlined below:
>
> **1. Motivation:**
> Prior works [1][2] focus on the multimodal hallucination task, [1] by enhancing visual attention, and [2] by subtracting visual logits to counteract language priors. In contrast, our motivation is orthogonal: we aim to reduce image interference via attention reallocation and fuse parametric and retrieved knowledge via logits addition to enhance knowledge utilization. This distinct objective is supported by thorough analysis and empirical validation, indicating that our method is not a simple reuse of existing techniques.
>
> **2. Method Design:**
> Unlike [1][2], which apply fixed factors for attention reweighting or logits subtraction, our method employs retrieval similarity and question-context similarity to dynamically adjust attention scores for image and context tokens. Additionally, we leverage these attention scores for adaptive knowledge fusion, enabling the model to selectively utilize different knowledge sources. This design is particularly beneficial when retrieved knowledge is noisy or partially incorrect.
>
> **3. Performance:**
> To further demonstrate the uniqueness and effectiveness of our approach, we conducted ablation studies replacing our components with those from [1][2]. As shown in the table below, these substitutions consistently led to performance drops, validating the advantage of our design over prior methods.
>
>
> | Model                      | InfoSeek | ViQuAE |
> |---------------------------|----------|--------|
> | Ours                      | 58.35    | 55.91  |
> | Ours replaced with [1]    | 54.13    | 53.25  |
> | Ours replaced with [2]    | 55.24    | 54.78  |
>
> We appreciate the reviewer’s comment and will include a more detailed discussion and comparison with related work in the revised version.
>
> ---
>
> **[W2]** Inference cost.
>
> **[R2]** We appreciate the reviewer’s concern regarding the inference cost. Compared with the baseline MRAG, our model introduces a modest overhead for parametric knowledge modelling. However, **the extra cost remains acceptable as the input for modeling parametric knowledge without context is relatively short**. Besides, **all compared methods except the baseline need to model both contextual and parametric knowledge.** The inference cost of our method is comparable to that of the compared methods in a broader context. We evaluate all compared methods over one sample on both discriminative and generative tasks. The tabulated results below confirm that our method takes comparable inference time. In fact, our method can achieve competitive inference efficiency as the baseline MRAG by processing the two knowledge sources in parallel. We will elaborate on this issue with the new experiments in the revised manuscript.
>
> | Model   | InfoSeek          | Human            |
> |---------|-------------------|------------------|
> | MRAG    | 0.46s (1.00x)     | 0.50s (1.00x)    |
> | CD      | 0.62s (1.35x)     | 0.72s (1.44x)    |
> | CAD     | 0.62s (1.34x)     | 0.72s (1.44x)    |
> | AdaCAD  | 0.62s (1.35x)     | 0.73s (1.45x)    |
> | Entropy | 0.63s (1.37x)     | 0.73s (1.46x)    |
> | COIECD  | 0.62s (1.35x)     | 0.73s (1.45x)    |
> | VCD     | 0.63s (1.38x)     | 0.74s (1.47x)    |
> | AGLA    | 0.83s (1.81x)     | 1.02s (2.04x)    |
> | Ours    | 0.62s (1.35x)     | 0.73s (1.46x)    |
>
> ---
>
> References
>
> [1] Paying More Attention to Image: A Training-Free  Method for Alleviating Hallucination in LVLMs.
>
> [2] Mitigating Object Hallucinations in Large Vision-Language Models through Visual Contrastive Decoding.

---

> > ### Comment · Reviewer_NFNU · 2025-08-06
> > **Rebuttal Acknowledged**
> >
> > Thanks for authors' detailed and timely rebuttal. All of my questions have been answered with very impressive answers and qualitative results, and I am happy to raise my score :").

---

> > > ### Author Response · Authors · 2025-08-07
> > >
> > > Dear Reviewer NFNU,
> > >
> > > We sincerely thank you for reading our rebuttal and for recognizing the value of our work. We greatly appreciate your constructive feedback and will ensure that the corresponding analyses and efficiency experiments are incorporated into the revised manuscript.
> > >
> > > Authors of Submission 14912

---

### Note · Authors · 2025-08-12

We sincerely thank you for your thoughtful and constructive feedback and for recognizing the improvements and achievements we have made. The reviewers' comments are extremely valuable to us. They not only fully affirm the existing strengths of our work but also provide important inspiration and direction for further improvement. We believe that these suggestions will significantly enhance the rigor, clarity of presentation, and overall academic impact of this study.

---

### Decision · Program_Chairs · 2025-09-17

**Decision:**

Accept (oral)

**Comment:**

This work introduces ALFAR, a training-free and plug-and-play method to enhance Multimodal Large Language Models (MLLMs) in knowledge-intensive tasks by better utilizing retrieved contextual knowledge. The approach addresses two key challenges: attention bias toward different tokens and conflicts between parametric and contextual knowledge. ALFAR improves model reasoning by (1) reallocating attention from visual to context tokens based on query relevance, and (2) adaptively fusing logits to balance parametric and retrieved knowledge. Extensive experiments across multiple MLLMs and benchmarks show that ALFAR consistently outperforms prior methods, achieving state-of-the-art performance without additional training or external tools. During the rebuttal, the authors successfully addressed all reviewer concerns, and we have decided to accept this work. We encourage the authors to incorporate the promised changes and refinements in the camera-ready version.